# High-Temperature Oxidation Behavior of Fe–10Cr Steel under Different Atmospheres

**DOI:** 10.3390/ma14133453

**Published:** 2021-06-22

**Authors:** Lei Cheng, Bin Sun, Chongyang Du, Wei Gao, Guangming Cao

**Affiliations:** 1School of Mechanical Engineering, Shenyang University, 21 Wanghua South Street, Shenyang 110044, China; chenglei@syu.edu.cn (L.C.); duchongyang@syu.edu.cn (C.D.); 2School of Materials Science and Engineering, Sun Yat-sen University, 135 Xingang West Road, Guangzhou 510000, China; gaowei@sysu.edu.cn; 3State Key Laboratory of Rolling and Automation, Northeastern University, 11 Wenhua Road, Shenyang 110819, China; caogm@ral.neu.edu.cn

**Keywords:** Fe–10Cr steel, Cr oxide, oxidation kinetics, oxide scale

## Abstract

Using a thermogravimetric analyzer (TGA), Fe–10Cr steel was oxidized in dry air and in a mixed atmosphere of air and water vapor at a relative humidity of 50% and a temperature of 800–1200 °C for 1 h. The oxidation weight gain curves under the two atmospheres were drawn, the oxidation activation energy was calculated, and the phase and cross-sectional morphology of the iron oxide scales were analyzed and observed by X-ray diffractometry (XRD) and optical microscopy (OM). The results showed that when the oxidation temperature was 800 °C, the spheroidization of Fe–10Cr steel occurred, and the oxidation kinetics conformed to the linear law. At 900–1200 °C, the oxidation kinetics followed a linear law in the preliminary stage and a parabolic law in the middle and late stages. In an air atmosphere, when the oxidation temperature reached 1200 °C, Cr_2_O_3_ in the inner oxide layer was partially ruptured. In an atmosphere with a water vapor content of 50%, Cr_2_O_3_ at the interface reacted with H_2_O to generate volatile CrO_2_(OH)_2_, resulting in a large consumption of Cr at the interface. At the same time, a large number of voids and microcracks appeared in the iron oxide layer, which accelerated the entry of water molecules into the substrate, as well as the oxidation of Fe–10Cr steel, and caused the iron oxide scales to fall off. Due to the volatilization of Cr_2_O_3_ and the conversion from internal oxidation to external oxidation, the internal oxidation zone (IOZ) of Fe–10Cr steel under water vapor atmosphere decreased or even disappeared.

## 1. Introduction

Steel has many special properties and irreplaceable advantages and has always been one of the most important materials in industrial production [1]. Strip steel accounts for a large proportion of steel products, and with the development of economy and society, the proportion of strip steel will continue to increase. The production process of hot-rolled strip includes heating in a heating furnace, high-pressure water descaling, rough rolling and finishing rolling, cooling, and coiling. The temperature of strip steel in hot rolling is usually as high as 800–1200 °C. Due to exposure to a mixed atmosphere of high-temperature air and water vapor, the surface of the strip steel will be seriously oxidized and produce a certain thickness of iron oxide scales, which will affect the surface quality and corrosion resistance of strip steel products [2,3]. Data show that the iron oxide scales produced during hot rolling, quenching, and tempering heat treatment of steel involve 7–10% of the total steel, which causes huge waste of resources and loss of corporate profits [4].

In order to reduce iron oxide scales production in the hot rolling process and improve the oxidation resistance of steel, research has been carried out on heating systems, rolling systems, and alloying elements [5,6,7,8]. It has been widely recognized that adding Cr element to steel can improve the oxidation resistance of steel. Therefore, many researchers have conducted extensive research on the oxidation behavior of Cr-containing steel at high temperature [9,10,11,12]. Yang [13] studied the oxidation behavior of low-carbon steel with a Cr content of 0.2% and of stainless steel with a Cr content of 15% in air. The results showed that after high-temperature oxidation, the microstructure of iron oxide scales in carbon steel and stainless steel was significantly different. The iron oxide layer of carbon steel was mainly a typical three-layer structure, that is, an outermost layer of Fe_2_O_3_, a middle layer of Fe_3_O_4_, and an innermost layer of FeO [14]. The structure of the iron oxide layer of stainless steel was mainly composed of Fe–Si spinel, Fe–Cr spinel, and Cr_2_O_3_. At the same temperature and atmosphere, the thickness of the oxide scales in carbon steel was about three times that of stainless steel. Cheng [15] found that when 21Cr–0.6Mo–Ni–Ti ferritic stainless steel was oxidized at high temperature, i.e., at 1100 °C, in air and water vapor atmosphere, oxide scales on the steel surface were composed of Fe–Cr spinel in the outer layer and Cr_2_O_3_ in the inner layer in both atmospheres. Due to the presence of water, the thickness of the oxide scales in the steel was greater than that produced in air only. The control of thickness and structure of the iron oxide scales is a key process for the surface quality control of strip steel [16]. Cr_2_O_3_ and Fe–Cr spinel produced by Cr-containing alloys at high temperature are the main responsible for a high oxidation resistance. Cr_2_O_3_ and Fe–Cr spinel can increase the activation energy of metal ion diffusion and slow down the diffusion rate of metal ions at the interface, thereby reducing the growth of oxide scales and achieving the purpose of improving the oxidation resistance of the steel [17]. The material of the roll is also an important factor affecting the surface quality of strip steel. In industrial production, the surface quality of strip steel is improved by using high-speed steel rolls instead of rolls made of high-chromium cast iron and by using low-pressure and high-water cooling methods to reduce oxide film flaking on the surface of rolls [18]. Currently, most research on Cr-containing steels is focused on ordinary structural steels with low Cr content and on stainless steels and heat-resistant steels with high Cr content. Apart from studies on ordinary structural steel and stainless steel, there are few reports on high-temperature oxidation of Cr-containing steel. A lower Cr content will result in poorer oxidation resistance. If the Cr content in the steel is high, it will produce a harmful σ-phase in the steel, thereby reducing steel toughness [19].

In this study, we focused on the oxidation kinetics, cross-sectional morphologies, and structures of the iron oxide scales of steel plates with a Cr content of 9.88% in a dry air and in a mixed gas atmosphere of air with 50% relative humidity and water vapor. The growth mechanism of the oxide scales on steel under air and water vapor atmosphere is discussed. By simulating the growth process of the iron oxide scales on steel surface in the actual hot rolling production process, we provide a theoretical basis for understanding high-temperature oxidation and hot rolling production of Cr-containing steel.

## 2. Experiment

The material used in this experiment was melted in a melting furnace, and its Cr content was 9.88% (herein, collectively referred to as Fe–10Cr steel). The chemical composition is shown in Table 1.

Using a WEDM (wire electrical discharge machining, Sanguang Machine Factory, Taizhou, China), the steel plate was cut into several specimens with a size of 10 mm × 8 mm × 2.5 mm, and a small hole with a diameter of 1 mm at the upper end of the specimen was drilled to suspend the specimen in the furnace chamber of a thermogravimetric analyzer (TGA). The specimens were polished step by step with SiC sandpapers (200#, 400#, 600#, 800#, 1000#, 1200#, and 1500#) until there no macro defects were present on the surface, then were placed in acetone for ultrasonic cleaning for 15 min to remove grease on the surface, put in alcohol for ultrasonic cleaning for 15 min, and finally dried using a blower.

In order to obtain complete oxidation weight gain data, high-temperature oxidation experiments were carried out in the simultaneous TGA S60/58507 manufactured by SETARAM in Lyon, France. First, the specimen was suspended in the furnace chamber. In order to avoid effects of the original atmosphere in the furnace on the experimental results, first, high-purity argon gas was injected into the furnace at a flow rate of 20 mL/min for 30 min. Then, the temperature in the chamber was increased to the target temperature (800 °C, 900 °C, 1000 °C, 1100 °C, and 1200 °C) at a heating rate of 30 °C/min. At this time, the argon gas feed tube was closed, and dry air and mixed gas containing 50% water vapor were respectively fed into the furnace chamber at a flow rate of 50 mL/min for oxidation for 1 h. The specific parameters of oxidation are shown in Table 2. During the oxidation process, the electronic balance attached to the TGA recorded the mass change of the specimen in real time. After the oxidation experiment, argon gas was introduced into the furnace chamber at a flow rate of 100 mL/min, while the specimens were lowered to room temperature at a rate of 40 °C/min. After the experiment, the specimens were mounted with a ZXQ-5 automatic thermal mounting machine (Laizhou Weiyi Experiment Machine Manufacturer, Laizhou, China) at a pressure of 27 Mpa and a temperature of 150 °C. Then, the specimens were sanded with 800 to 1500# SiC sandpapers and polished with 2.5# and 1.5# diamond polishing paste, respectively. X-ray diffractometry (XRD) was used to analyze the composition and structure of the oxides (Cu-K_α_, 2θ = 20°−120°, step size was 0.02°, step rate was 1.2°/min). An optical microscope (OM) was used to observe the cross-sectional morphology of the oxides (OLYMPUS-BX53M, Olympus (China) Co. Ltd, Beijing, China).

## 3. Results and Discussion

### 3.1. Oxidation Kinetics

Figure 1a,b show the oxidation weight gain of Fe–10Cr steel under two atmospheres. It can be seen from Figure 1 that the oxidation weight gain of Fe–10Cr steel under water vapor atmosphere was significantly greater than that under air at the same oxidation temperature. Under the same oxidizing atmosphere, the oxidation weight gain of Fe–10Cr steel increased with the increase of the oxidizing temperature. When the oxidation temperature was 800 °C, the oxidation weight gain of Fe–10Cr steel under both oxidizing atmospheres was small, while when the temperature rose to 1200 °C, the oxidation weight gain of Fe–10Cr steel was significantly higher than at the other temperatures.

### 3.2. XRD Test Results

Figure 2 shows the X-ray diffraction patterns of Fe–10Cr steel oxidized in dry air for 1 h. XRD only detected Fe in Fe–10Cr steel after oxidation at 800 °C for 1 h. When the oxidation temperature was 900 °C, 1000 °C, 1100 °C, and 1200 °C, the XRD detected Cr_2_O_3_, Fe_3_O_4_, and Fe_2_O_3_, whereas no FeO was detected.

Figure 3 shows the XRD test results for Fe–10Cr steel oxidized for 1 h in a water vapor atmosphere with a relative humidity of 50%. When the temperature was 800 °C, the phases of the iron oxide scales were Fe and Fe_3_O_4_, respectively. When the temperature rose to 900 °C, XRD detected Fe_2_O_3_, Fe_3_O_4_, FeO, Fe_2_SiO_4_, FeCr_2_O_4_, and Cr_2_O_3_. When the oxidation temperature was 1000 °C, Fe_2_O_3_, Fe_3_O_4_, and FeO were detected in the iron oxide scales. When the temperature rose to 1100 °C and 1200 °C, only Fe_2_O_3_ was detected in the iron oxide layer.

By comparing the XRD results of Fe–10Cr steel oxidized in the two experimental atmospheres, it appeared that when Fe–10Cr steel was oxidized in an atmosphere containing water vapor, the phase change in the oxide scale layer was more complex than in dry air.

### 3.3. Cross-Sectional Morphology of Oxide Scales

Table 3 shows the cross-sectional morphology of the iron oxide scales of Fe–10Cr steel oxidized in the two atmospheres. When oxidized in dry air at 800 °C, only a small number of oxidized nodules nucleated on the surface of the substrate. The gray oxide near the substrate was Fe_3_O_4_, and the side portion near the inlay material was a thin layer of Fe_2_O_3_. At 900 °C, from the inlay material to the side of the substrate, a continuous and thick layer of iron oxide scales was formed on the surface of the substrate that could be divided into three parts: an outer oxide layer, an inner oxide layer, and an inner oxide zone. According to the XRD test results, the outer oxide layer included Fe_2_O_3_ and Fe_3_O_4_, the inner oxide layer was a mixture of Fe_3_O_4_ and Cr_2_O_3_, and the inner oxide zone was a very thin layer of dotted Cr_2_O_3_. At 1000 °C, the particle size and aggregation degree of Cr_2_O_3_ in the inner oxide zone were greatly increased than at 900 °C, and the inner oxide layer was partially aggregated into stripes. At 1100 °C, the particle size of internal oxide continued to increase. When the oxidation temperature was 1200 °C, holes appeared in the iron oxide layer. Meanwhile, the size of dotted Cr_2_O_3_ in the internal oxidation zone tended to decrease towards the side of the substrate.

When Fe–10Cr steel was oxidized in a mixed atmosphere of air and water vapor with a relative humidity of 50% at 800 °C, only a part of the nodular oxide nucleated on the surface of the substrate, and the side portion close to the substrate was composed gray Fe_3_O_4_. Only a small amount of Fe_2_O_3_ was generated on the side close to the inlay material. When the temperature rose to 900 °C, the substrate was covered by iron oxide scales, which included an outer oxide layer composed of iron oxide and an inner oxide layer composed of chromium oxide. Combined with the XRD test results, these findings indicate that the outer oxide layer was mainly composed of Fe_2_O_3_, Fe_3_O_4_, and FeO, and the inner oxide layer mainly consisted of a mixture of FeCr_2_O_4_ and FeO, with Cr_2_O_3_ near the interface of the substrate. When the oxidation temperature reached 1000 °C, the outer oxide layer was mainly iron oxide, and the inner oxide layer was mainly chromium oxide. When the temperature was 1100 °C and 1200 °C, the thickness of the iron oxide scales increased compared to that at 1000 °C; the outer oxide layer was mainly composed of Fe_2_O_3_, and the inner oxide layer consisted of chromium. At 900 °C, 1000 °C, and 1100 °C, we observed a large number of voids and microcracks in the iron oxide layer. When the oxidation temperature was 1200 °C, the outer oxide layer displayed peeling of the iron oxide scales.

By comparing the oxidation cross-sectional morphology of Fe–10Cr steel, we found that under air conditions, no FeO formed in the iron oxide layer; FeO formed in the iron oxide layer of Fe–10Cr steel when oxidation occurred in a mixture of air and water vapor with a relative humidity of 50%. At the same oxidation temperature, the thickness of the Fe_2_O_3_ layer in Fe–10Cr steel under dry air conditions was greater than that under water vapor conditions.

### 3.4. Iron Oxide Scale Thickness

Figure 4 shows the thickness of the iron oxide scales of Fe–10Cr steel after oxidation in air and water vapor atmospheres for 1 h. At 800 °C, Fe–10Cr steel spheroidized in both oxidizing atmospheres, and no continuous iron oxide scale was formed. The thickness was extremely uneven and could not be measured. As the temperature continued to rise, the iron oxide scales on the surface of the substrate also thickened. At all oxidation temperatures, the thickness of the iron oxide scales of Fe–10Cr steel produced in water vapor atmosphere was greater than that in air conditions, in agreement with the oxidation weight gain curve in Figure 1. Under air conditions, when the oxidation temperature was between 900 and 1100 °C, the thickness of the iron oxide scales increased slowly, and when the temperature rose to 1200 °C, the thickness of the iron oxide scales increased greatly. When oxidized in a water vapor atmosphere, the thickness of the iron oxide scales of Fe–10Cr steel also increased significantly.

### 3.5. Oxidation Kinetic Model

In a preliminary stage, when Fe–10Cr steel is directly exposed to the oxidizing atmosphere, oxygen molecules collide with the substrate and are adsorb on the surface of the substrate through van der Waals forces. O_2_ is decomposed into O^2−^ and interacts with free electrons in the substrate allowing chemical adsorption. As the oxidation time increases, iron oxides nucleate and grow on the surface of the substrate, eventually forming continuous iron oxide scales. This stage is mainly a ‘gas–solid’ reaction, and the reaction rate is relatively fast. At this time, the interface reaction is the controlling step. The oxidation of the alloy at this stage follows a linear law [19], that is, the oxidation weight gain and the oxidation time of the specimen conform to the relationship of Equation (1):(1)$$\Delta W=K_{1}t$$
where Δ*W* is the oxidation weight gain per unit area (mg/mm^2^), *K*_1_ is the linear constant (mg/mm^4^·s), and *t* is the oxidation time (s).

According to Equation (1), the linear rate constant of Fe–10Cr steel in the preliminary stage can be determined and is shown in Table 4.

According to Table 4, the K_1_ values of Fe–10Cr steel in water vapor atmosphere were greater than those in air atmosphere. As the oxidation temperature increased, the *K*_1_ values also increased. It can be seen from Table 3 that when the oxidation temperature was 800 °C, Fe–10Cr steel produced only a part of nodular oxides in both air and water vapor atmospheres and did not produce continuous iron oxide scales. Therefore, it can be stated that the oxidation behavior of Fe–10Cr steel at 800 °C within the target time conforms to the linear law, but it is predictable that, as the oxidation time continuous to increase, Fe–10Cr steel will also produce continuous iron oxide scales at 800 °C.

When a continuous thick iron oxide is generated on the surface of a substrate, the substrate and the air become separated by the iron oxide layer. At this time, air needs to pass through the iron oxide layer to further oxidize the substrate. This process includes a reaction between the air and the surface of the iron oxide scales, a reaction between the interface of the iron oxide scale and the substrate, and a bidirectional diffusion of iron and oxygen ions. As the oxidation time increases, the iron oxide scales continue to thicken, and the diffusion path of iron and oxygen ions gradually becomes longer. Therefore, in the middle and late stages of the oxidation of Fe–10Cr steel, the oxidation rate begins to slow down, and the oxidation kinetics conforms to the parabolic law, that is, the oxidation weight gain and oxidation time satisfy the Kofstad [20] Equation:(2)$${\left(\Delta W\right)}^{2}=K_{p}t$$
where Δ*W* is the oxidation weight gain per unit area (mg/mm^2^), *K_p_* is the parabolic constant (mg/mm^4^·s), and *t* is the oxidation time (s).

According to Equation (2), the *K_p_* value of Fe–10Cr steel in two atmospheres is calculated, as shown in Table 5.

According to the derivation of Arrhenius and Neviobalo [21], the relationship between the oxidation rate constant of steel and the oxidation activation energy is as follows: (3)$$K_{p}=K_{0}\cdot \exp(-Q/RT)$$
where *K*_0_ is the model constant, *Q* is the activation energy of the alloy (J/mol), *T* is the oxidation temperature (*K*), and *R* is the gas constant (8.314 J/(mol·K)).

Considering the logarithm of both sides of Equation (3), we obtain:(4)$$\ln K_{p}=\ln K_{0}+(-Q/RT)$$

In the experiment, the *K_p_*, *R*, and *T* values of the steel grade are known, and the values of ln*K_p_* and 1/*T* can be obtained by calculation; the calculation results of Equation (4) are plotted and fitted to obtain the fitted values. Through the fitted straight line of ln*K_p_* and 1000/*T*, the activation energy of Fe–10Cr steel under the two oxidizing atmospheres could be obtained from the slope of the straight line (-Q/RT). Figure 5 shows the fitting diagram of Fe–10Cr steel at 900–1200 °C. The oxidation activation energy of Fey-10Cr steel in air resulted to be 311.4 kJ/mol, and the activation energy in a mixed atmosphere with a water vapor content of 50% was 204.6 kJ/mol. Because the oxidation activation energy characterizes the height of the energy barrier that needs to be crossed during oxidation, it can also explain how difficult an oxidation process can be. Therefore, when Fe–10Cr steel was oxidized in an atmosphere containing water vapor, the oxidation process could proceed easily, and steel was oxidized and corroded more severely.

### 3.6. Spheroidization of Fe–10Cr Steel

When the temperature was 800 °C, spherical oxides formed in both oxidizing atmospheres. This is the ‘nodulation’ phenomenon that occurs during high-temperature oxidation of metals [22,23] and is usually related to the stress of the metal oxide during the growth process. Since the volume of the oxide generated per unit volume of metal after oxidation is often greater than 1, as the iron oxide continues to grow, growth stress will be generated in the iron oxide scales. When the growth stress is large enough and cannot be released by plastic deformation, the oxide scales will crack, and the cracks will penetrate the oxide scales [19]. At this time, the oxidizing atmosphere quickly penetrates through the cracks along the metal grain boundaries and reacts rapidly deep in the substrate, thereby forming nodular oxides. The schematic diagram of nodule formation of Fe–10Cr steel is shown in Figure 6.

### 3.7. Oxidation Mechanism of Fe–10Cr Steel in Air

When Fe–10Cr steel was oxidized in air, in the preliminary stage of oxidation, due to the high partial pressure of oxygen in air, iron oxides first nucleated and grew on the surface of the substrate. Because the interface reaction dominated at this stage, the speed was extremely high. According to the Pt labeling experiment conducted by Fukumoto [24], when an alloy is oxidized, the growth of the outer oxide layer is dominated by the outward diffusion of iron, while the growth of the inner oxide layer is mainly caused by the inward diffusion of oxygen. The ΔG^0^–T diagram drawn by Ellingham shows that the affinity between chromium and oxygen is greater than that between iron and oxygen [25]; therefore, as oxygen diffuses inward, the Cr element in the steel will selectively be oxidized by oxygen to form dot-shaped or strip-shaped oxide Cr_2_O_3_ scattered between the substrate and the inner oxide layer, and an internal oxidation zone (IOZ) is formed. As the oxidation continues, FeO will wrap Cr_2_O_3_ and revert to the solid phase. The reaction formula is shown in Equation (5). FeCr_2_O_4_ was not detected during XRD analysis, which may be due to a small amount of FeCr_2_O_4_ in the oxide scales, whose peaks in the XRD diagram were covered by the other larger peaks. At this time, Fe–10Cr steel had completed the process from internal oxidation to external oxidation. As oxygen continued to diffuse deeper into the substrate, an IOZ was regenerated, then the conversion from internal oxidation to external oxidation was completed again, the iron oxide continued to thicken, and the IOZ continued to expand to the side of the substrate.
(5)$$\mathrm{FeO}+{\mathrm{Cr}}_{2}O_{3}={\mathrm{FeCr}}_{2}O_{4}$$

Although Fe–10Cr steel did not produce a continuous and dense Cr_2_O_3_ layer when oxidized in air, it largely hindered the two-way diffusion of iron and oxygen ions, especially the outward diffusion of iron ions. Therefore, under air conditions and at 900–1100 °C, the thickness of the Fe_2_O_3_ layer in the outer layer of the iron oxide scales occupied a large proportion in the entire iron oxide layer, which indicates that the formed Cr_2_O_3_ played contributed to increasing the oxidation resistance of the steel. After observing the cross-sectional morphology of oxidation in air at 1200 °C, a large number of holes appeared in the inner oxide layer, and the thickness of the Fe_2_O_3_ layer in the entire iron oxide layer decreased proportionally. This phenomenon indicates that the inner oxide layer of Fe–10Cr steel was partially ruptured due to the increase of the oxidation temperature. That is, at high temperature, Cr_2_O_3_ in the inner oxide layer reacted with O_2_ to generate CrO_3_ gas that was volatilized outward, generating holes. The reaction schematic diagram is shown in Figure 7, and the reaction formula can be found in Equation (6). Due to the partial failure of the inner oxide layer, the protective Cr_2_O_3_ was reduced, and the presence of holes provided a favorable channel for the two-way diffusion of iron and oxygen ions, thereby reducing the proportion of Fe_2_O_3_ in the iron oxide layer and accelerating the oxidation rate of steel. Therefore, the thickness of the iron oxide scale increased greatly at 1200 °C.
(6)$${\mathrm{Cr}}_{2}O_{3}\left(s\right)+{3/2O}_{2}\left(g\right)={2\mathrm{CrO}}_{3}\left(g\right)$$

### 3.8. Oxidation Mechanism of Fe–10Cr Steel under Water Vapor Conditions

When Fe–10Cr steel was oxidized in a mixed atmosphere containing 50% water vapor, a large number of voids and microcracks formed in the iron oxide layer, and the oxide scales were thicker than under air conditions. The reasons for these defects are as follows. The water vapor in the oxidizing atmosphere diffused inward and reacted with Cr oxide in the iron oxide layer to produce the volatile hydroxide CrO_2_(OH)_2_; the reaction formula corresponds to Equation (7). When CrO_2_(OH)_2_ rapidly diffused outward, a large number of holes appeared in the iron oxide layer. These holes provided ion vacancies and electron holes for the diffusion of iron ions and electrons, thereby accelerating the diffusion rate of iron ions and electrons and speeding up the oxidation rate of Fe–10Cr steel. At the same time, the presence of water vapor accelerated the growth of iron oxide scales; therefore, growth stress in the iron oxide layer also increased sharply, thereby promoting the formation of microcracks. In the atmosphere, water molecules invaded the substrate through microcracks and reacted to form iron oxide and H_2_. The generated H_2_ passed through the microcracks in the reverse direction and reacted with the iron oxide in the iron oxide layer to form H_2_O. The presence of microcracks and H_2_ also explain the decrease in the proportion of Fe_2_O_3_ layer in the iron oxide scale layer and the presence of FeO. The H_2_O generated by the reaction invaded the substrate through the microcracks again; therefore, the microcracks became the diffusion channels of H_2_ and H_2_O, the oxide scales continued to grow, and the substrate continued to be corroded. Due to the existence of holes and microcracks under water vapor, the oxidation rate of Fe–10Cr steel continued to increase, the morphology of the iron oxide scales also changed significantly, and even spalling occurred.
(7)$${1/2\mathrm{Cr}}_{2}O_{3}\left(s\right)+H_{2}O(g)+{3/4O}_{2}\left(g\right)={\mathrm{CrO}}_{2}{\left(\mathrm{OH}\right)}_{2}\left(g\right)$$

By comparing the cross-sectional morphology of Fe–10Cr steel in air and water vapor atmospheres, it can be seen that under air conditions, a large number of dot-like and strip-like internal oxides at the interface between the substrate and the iron oxide scales formed, that is, an IOZ was present, and the IOZ rarely disappeared under water vapor conditions. Xu [26] found that when Fe–20Cr steel was oxidized in air at 800 °C, the cross-section of Fe–20Cr steel presented only a dense Cr_2_O_3_ layer. Cheng [27] investigated the oxidation behavior of Fe–16Cr steel at 1000–1150 °C under a water vapor atmosphere with a relative humidity of 18%. The results showed that with the participation of water vapor, a dense Cr_2_O_3_ layer formed in Fe–16Cr steel was destroyed, the Cr element at the interface was heavily depleted, and the IOZ showed dot and strip internal oxides. Studies have shown [28] that in Fe–Cr alloys, when the Cr content is greater than 14%, a complete Cr_2_O_3_ layer forms on the surface of the alloy at the initial stage of oxidation to prevent the two-way diffusion of iron and oxygen ions, thus achieving oxidation resistance. When high chromium steel is oxidized in air, it will form a complete and dense Cr_2_O_3_ layer, which hinders the formation of iron oxides. When it is oxidized in water vapor atmosphere, the dense Cr_2_O_3_ generated in the steel undergoes the reaction of Equation (7), and a large amount of Cr elements volatilizes. At this time, fracture oxidation occurs, that is, the outer oxide layer begins to form from iron oxide. The destroyed Cr_2_O_3_ layer is wrapped by the iron oxide layer and undergoes a solid-phase reaction to form an inner oxide layer mainly composed of Fe–Cr spinel. Figure 8 shows a schematic diagram of the Fe–10Cr oxide layer. For the Fe–10Cr steel used in this experiment, when it was oxidized in air, due to the conversion from internal oxidation to external oxidation, the IOZ continued to migrate to the side of the substrate. When it was oxidized in a water vapor atmosphere with a relative humidity of 50%, a large amount of Cr_2_O_3_ at IOZ volatilized. At the same time, under the combined action of the two, a transformation from internal oxidation to external oxidation occurred, and the IOZ decreased or even disappeared.

## 4. Conclusions

(1)After being oxidized in air for 1 h, Fe–10Cr steel scales include three parts: an outer oxide layer, an inner oxide layer, and an IOZ. The outer oxide layer is Fe_2_O_3_ and Fe_3_O_4_, the inner oxide layer is a mixture of Fe_3_O_4_ and Cr_2_O_3_, and the IOZ is dotted or striped Cr_2_O_3_. In water vapor atmosphere, the oxide scales are composed of two parts: an outer oxide layer and an inner oxide layer. The outer oxide layer is iron oxide, and the inner oxide layer is a product with chromium oxide and spinel structures.(2)At 800 °C, the oxidation kinetics of Fe–10Cr steel under the two atmospheres follows a linear law. The oxidation kinetics curve at 900–1200 °C follows a linear law in the preliminary stage and a parabolic law in the middle and late stages. The oxidation activation energy in air and water vapor is 311.4 kJ/mol and 204.6 kJ/mol, respectively. Fe–10Cr steel is more easily oxidized in water vapor, and oxidation is greater.(3)Under the action of growth stress, at 800 °C, Fe–10Cr steel will be spheroidized under the two atmospheres.(4)When oxidized in dry air at 1200 °C, the inner oxide layer of Fe–10Cr steel will be partially damaged, which will result in a significant increase in the thickness of the iron oxide scales and a decrease in the thickness of the Fe_2_O_3_ layer.(5)When oxidized in a mixed atmosphere containing 50% water vapor, a large number of voids and microcracks are generated in the iron oxide layer of Fe–10Cr steel; due to the existence of voids and microcracks, the oxidation rate of Fe–10Cr steel is accelerated, and the thickness of Fe_2_O_3_ decreases. Under the combined action of volatilization and internal and external oxidation, the IOZ of Fe–10Cr steel decreases or even disappears.

## Figures and Tables

**Figure 1 materials-14-03453-f001:**
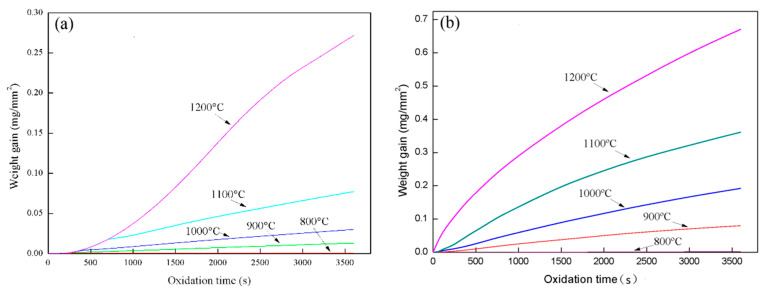
Oxidation weight gain curves of Fe–10Cr steel under different atmospheres: (**a**) air; (**b**) air with 50% water vapor.

**Figure 2 materials-14-03453-f002:**
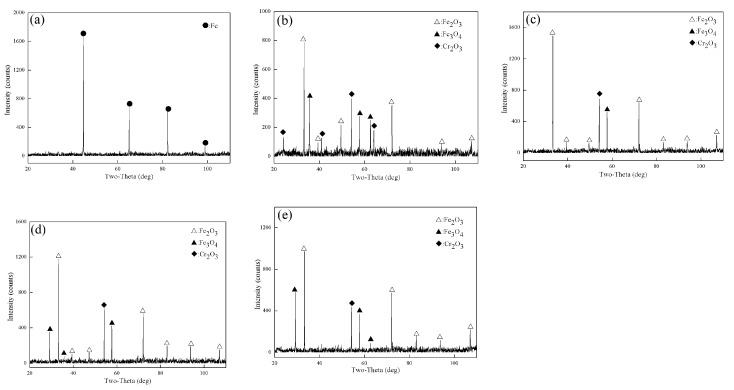
XRD patterns of Fe–10Cr steel in air atmosphere. (**a**) 800 °C; (**b**) 900 °C; (**c**) 1000 °C; (**d**) 1100 °C; (**e**) 1200 °C.

**Figure 3 materials-14-03453-f003:**
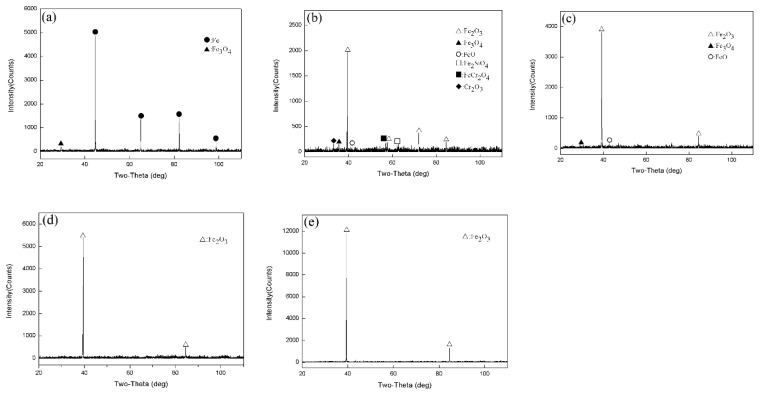
XRD patterns of Fe–10Cr steel in 50% water vapor. (**a**) 800 °C; (**b**) 900 °C; (**c**) 1000 °C; (**d**) 1100 °C; (**e**) 1200 °C.

**Figure 4 materials-14-03453-f004:**
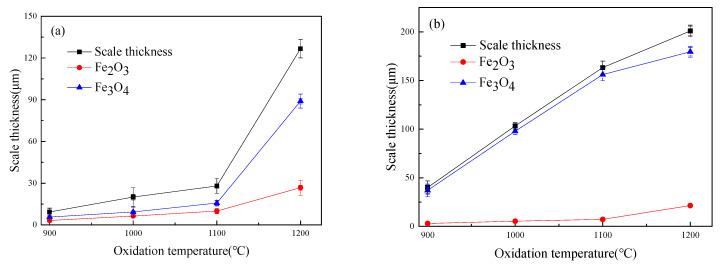
Oxide scale thickness of Fe–10Cr steel in different atmospheres. (**a**) Air; (**b**) air + 50% water vapor.

**Figure 5 materials-14-03453-f005:**
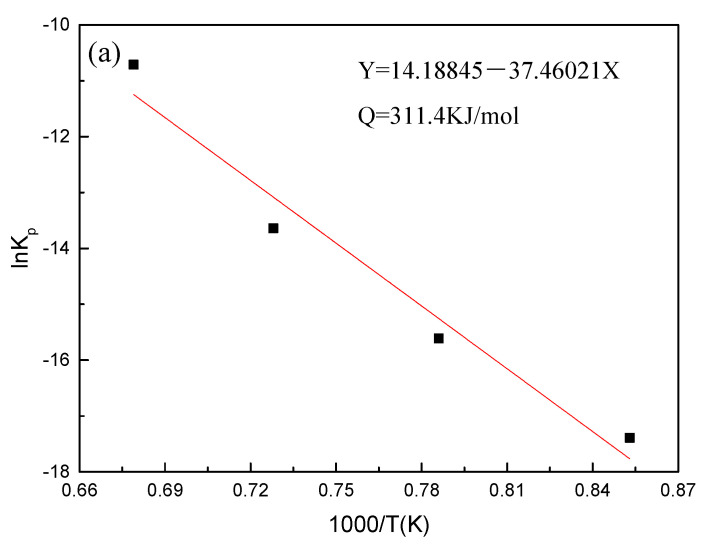
Fitting curve of ln*K*_p_ and 1000/*T* of Fe–10Cr steel in different atmospheres. (**a**) Air; (**b**) air + 50% water vapor.

**Figure 6 materials-14-03453-f006:**
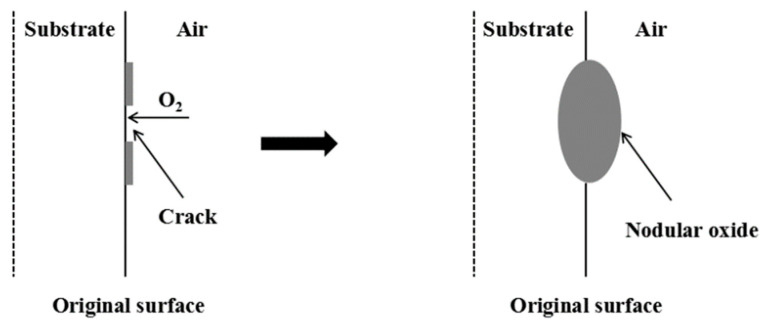
Schematic diagram of nodule formation of Fe–10Cr steel.

**Figure 7 materials-14-03453-f007:**
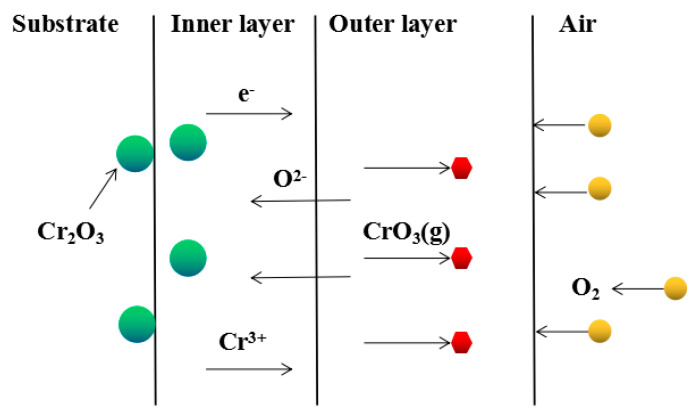
Schematic diagram of Cr_2_O_3_ failure in Fe–10Cr steel at 1200 °C in air.

**Figure 8 materials-14-03453-f008:**
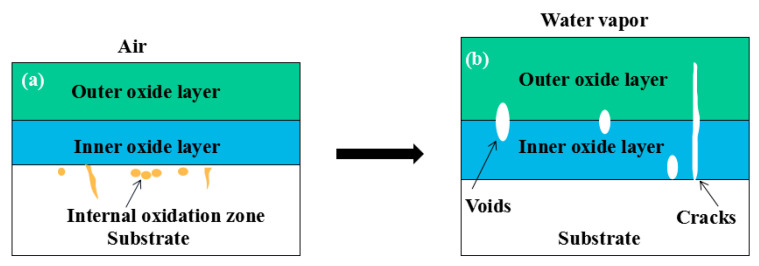
Schematic diagram of the oxide layer of Fe–10Cr steel in (**a**) air; (**b**) air + 50% water vapor.

**Table 1 materials-14-03453-t001:** Chemical composition of Fe-10Cr steel (wt%).

Sample	C	Si	Mn	P	S	Cr	Fe
Fe–10Cr steel	0.094	0.23	0.51	0.07	0.01	9.88	Bal

**Table 2 materials-14-03453-t002:** Oxidation experiment parameters.

Sample	Oxidation Time (h)	Temperature (℃)	Atmosphere
1	1	800, 900, 1000, 1100, 1200	Air
2	1	800, 900, 1000, 1100, 1200	Air + 50% water vapor

**Table 3 materials-14-03453-t003:** Cross-sectional morphology of Fe–10Cr steel oxidized in different atmospheres.

Oxidation Temperature	Air	Air + 50% Water Vapor
800 °C	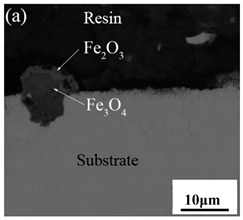	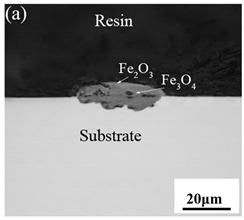
900 °C	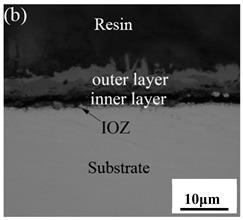	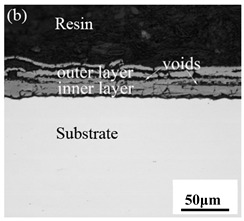
1000 °C	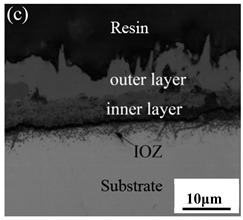	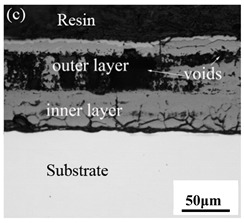
1100 °C	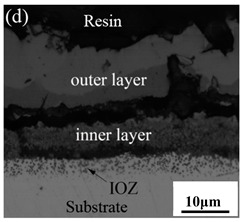	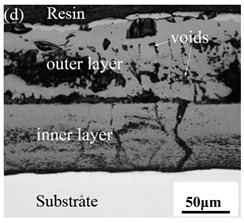
1200 °C	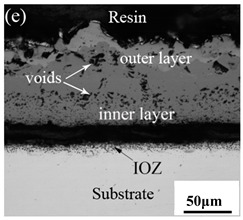	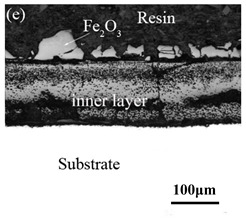

**Table 4 materials-14-03453-t004:** K_1_ values of Fe–10Cr steel in air and 50% water vapor.

Temperature/°C		800	900	1000	1100	1200
K_1_/(mg/mm^4^·s)	Air	3.040 × 10^−8^ (0–3600 s)	3.508 × 10^−6^ (0–1500 s)	8.595 × 10^−6^ (0–1100 s)	9.918 × 10^−6^ (0–800 s)	3.840 × 10^−5^ (0–500 s)
	Air + 50% Water vapor	4.906 × 10^−7^ (0–3600 s)	2.454 × 10^−5^ (0–800 s)	5.427 × 10^−5^ (0–550 s)	1.257 × 10^−4^ (0–400 s)	4.670 × 10^−4^ (0–200 s)

**Table 5 materials-14-03453-t005:** *K_p_* values of Fe–10Cr steel in air and 50% water vapor.

Temperature/°C		800	900	1000	1100	1200
K_p_/(mg/mm^4^·s)	Air	-	2.800 × 10^−8^	1.664 × 10^−7^	1.196 × 10^−6^	2.241 × 10^−5^
	Air + 50% Water vapor	-	1.374 × 10^−6^	9.122 × 10^−6^	3.182 × 10^−5^	1.036 × 10^−4^

## Data Availability

All data is contained within the article.

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
