# Peer review of "High-Temperature Oxidation Behavior of Fe–10Cr Steel under Different Atmospheres"

_materials, 2021, doi:10.3390/ma14133453_

Round 1

Reviewer 1 Report

please, see my comments in the enclosed file

The main topic/question addressed by the research is the high temperature corrosion problem for the steels, as it is well specified in the title which is in full accordance with the contents of the paper. The authors have well described the kinetic of this material degradation process, and their experiment has been developed into 2 different conditions, that is in wet and dry atmosphere. I can say that it has been interesting to follow the various stage of the corrosion but I cannot say that this is the only work that you can find about this specific issue. Nevertheless I can say that it is a good contribution to validate the state of the art, so that we can also consider the results of this work to improve our normative regarding the material. The paper is well written and the text is clear and easy to read, provided that my suggestions to improve it will be considered before the publication.

I have read the conclusions and I have found them consistent with the evidence and arguments presented. It is a summary of the work done. This is a kind of "conclusions" and I can accept them, but I think that the authors can improve the conclusions providing information for future development of this work.

Reviewer 2 Report

Paper deals with oxidation behaviour in various atmospheres in Cr alloyed steel. It has exciting potential; however, some issues need addressing, listed below.

  1. Some abbreviations are used without explanation. For example, IOZ, WEDM.
  2. In Introduction only light state of the art is given. There was a lot of work done on oxidation of Cr containing steel. I refer authors to papers dealing with rolls, where oxidation in high Cr rolls is important degradation mechanism.
  3. In Experimental methods it is not clear whether samples for XRD were mounted or not. Moreover, please give full experimental conditions for XRD, as step size, step rate and dwell time are missing.
  4. In Results section please investigate XRD data more thoroughly and try to quantify amount of each phase with separate thickness of layers presented in micrographs. It is also not clear how Fe2O3 and Fe3O4 were determined in micrographs. Reference for linear oxidation law is missing and should be given. Determined oxidation activation energy is given on 3 decimal places from linear fitting. As method itself produces uncertainty, more than 1 decimal place is not needed.
  5. Reader would appreciate some discussion where obtained results are compared to literature.
  6. In conclusion there is mistake in activation energy stated.
  7. References have "et al." at numerous entries, where all authors should be given.
  8. English language needs proper polish as some sentences are very difficult to understand.

Round 2

Reviewer 2 Report

Paper incorporated suggested changes. State of language is better, but still needs polishing. For example, "This may be due to the fact that FeCr2O4 is less and there are bigger peaks in the XRD diagram and FeCr2O4 is covered,so it cannot be detected by XRD."
